# OpenReview forum: "VERA: Variational Inference Framework for Jailbreaking Large Language Models"
_NeurIPS.cc/2025/Conference — NeurIPS 2025 poster_

### Official Review · Reviewer_heHX · 2025-06-06

**Clarity:** 3
**Significance:** 2
**Originality:** 3
**Rating:** 4
**Confidence:** 3

**Summary:**

This paper proposes VERA, a variational inference framework for attacking LLMs. VERA is a black-box attack method that requires API access of the target model. The authors compared the performance of VERA with a wide range of baselines including both white-box attack and black-box attack on both open-source model and closed source models. In addition, transferability and robustness to defenses are also studied in this paper.

**Questions:**

- Can you elaborate on the derivation of equation (7)? I did not follow this from (6) to (7). More explanations will be helpful.

- Experiments in Section 3.3 is a bit weird. The authors compare VERA with GPTFuzzer in detail in this section, why not in Section 4?

- Figure 2 (b) needs more explanations, especially the vertical lines in the plot, I am not following this part.

- Please add the simple baselines mentioned earlier.

- How the baseline results are collected? Is results in table 1 done on the same set of samples?

- For well safety-aligned models, any insights on how VERA will perform better?

I would love to increase my scores if the authors can address my main concerns.

**Ethical Concerns:**

["NO or VERY MINOR ethics concerns only"]

**Final Justification:**

The authors' rebuttal has addressed my main concerns by adding more ablation studies.

**Paper Formatting Concerns:**

The formatting looks good to me.

**Quality:**

2

**Strengths And Weaknesses:**

**Strengths**

- Formulating jailbreaking as variational inference seems interesting and relevant.

-  The experimental study in this work in comprehensive.

**Weakness**

- Even though the problem formulation is interesting, I am not sure how much of the attack performance is gained from the training given the authors provided a detailed system prompt (in the Appendix) and the target models are mostly not safely aligned (this method also has weaker performance on Llama-chat models).

- There are some simple yet straightforward baselines missing in this work, i.e., using the attacker model without training at all with the same system prompt. This will also address my first point as well.

---

> ### Author Rebuttal · Authors · 2025-07-31
>
> Thank you for the constructive review and for highlighting VERA’s formulation and comprehensive experimental study. We have addressed your concerns below.
> * **(W1)Even though the problem formulation is interesting, I am not sure how much of the attack performance is gained from the training given the authors provided a detailed system prompt (in the Appendix) and the target models are mostly not safely aligned (this method also has weaker performance on Llama-chat models).**
>
> Thank you for the thoughtful question. We conducted a new ablation to isolate the effect of training from the system prompt. Specifically, we compare VERA to a BoN baseline that uses the same system prompt but without any training (batch size 32):
>
> **Table 1: Effect of VERA training versus system‑prompt‑only BoN baseline.**
>
> | Method | VERA | BoN  |
> |--------|------|------|
> | ASR    | 94.0 | 48.5 |
>
> This large gap confirms that training is essential for VERA’s effectiveness, as the system prompt alone is insufficient.
> Regarding alignment, VERA performs well even against strong defenses. We evaluated on aligned models like R2D2 and LLaMA 3 (see Table 3) and against CircuitBreaker, where VERA maintains high ASR. While white-box methods sometimes achieve higher ASR, they are easier to detect (e.g., via perplexity filters), reducing their utility in real-world settings. VERA offers a stronger black-box alternative that remains effective across models with varying levels of alignment.
>
> * **(W2)There are some simple yet straightforward baselines missing in this work, i.e., using the attacker model without training at all with the same system prompt. This will also address my first point as well.**
>
> The to (W1) also addresses your concern in (W2): while the system prompt alone offers some attack capability, training is essential for reaching high performance.
>
> * **(Q1)Can you elaborate on the derivation of equation (7)? I did not follow this from (6) to (7). More explanations will be helpful.**
>
>
> The derivation follows from the standard REINFORCE gradient estimator (e.g., [3]), and we will include it in the revision to improve clarity. Specifically, our original objective in equation (4) is:
>
> $$
> E_{q_{\theta}}\left[\log P_{LM}(y^* \mid x) + \log P(x) - \log q_\theta(x)\right]
> $$
>
> Equation (6) defines the function inside the expectation as:
>
> $$
> f(x) = \log P_{LM}(y^* \mid x) + \log P(x) - \log q_\theta(x)
> $$
>
> Our goal is to compute the gradient of the expectation with respect to $\theta$:  $\nabla_{\theta} E_{q_\theta}[f(x)]$.
>
> Since the distribution $q_\theta$ depends on $\theta$, we apply the REINFORCE trick, which uses the identity:
> $$
> \nabla_\theta q_\theta(x) = q_\theta(x) \nabla_\theta \log q_\theta(x)
> $$
>
> Substituting into the gradient of the expectation:
>
> $$
> \nabla_\theta E_{q_\theta(x)} [f(x)] = \nabla_\theta \int f(x) q_\theta(x)  dx
> $$
>
> $$
> = \int f(x) \nabla_\theta q_\theta(x)  dx
> $$
>
> $$
> = \int f(x) q_\theta(x) \nabla_\theta \log q_\theta(x)  dx
> $$
>
> $$
> = E_{q_\theta} [f(x) \nabla_\theta \log q_\theta(x)]
> $$
> This yields equation (7), the REINFORCE gradient estimator. In practice, we estimate the expectation using Monte Carlo sampling:
>
> $$
> \nabla_\theta E_{q_\theta(x)} [f(x)] \approx \sum_i f(x_i) \nabla_\theta \log q_\theta(x_i), ~~x_i\sim q_\theta(x_i)
> $$
>
> This gradient encourages $q_\theta$ to place more mass on $x_i$ with high values of $f(x)$ and less on those with low values. As a result, the learned $q_\theta$ increasingly generates $x$ that produce higher $f(x)$. We visualize this in Figure 1, where the gradient updates shift $q_\theta$ toward regions where $f(x)$ is large.
>
>
> * **(Q2)Experiments in Section 3.3 is a bit weird. The authors compare VERA with GPTFuzzer in detail in this section, why not in Section 4?**
>
> We compare it to GPTFuzzer in Section 3.3 as it is a well-known template-based black-box attack, which is useful for analyzing prompt diversity, optimization behavior and scalability relative to VERA’s optimization-based approach. However, GPTFuzzer is not included in the HarmBench benchmark, which is why it was not part of the Section 4 evaluation originally.
>
> To address this gap and strengthen the analysis, we have now included AutoDAN alongside GPTFuzzer in Section 3.3, both methods are representative automated black-box attacks that operate using initial prompt templates. The results show the same trends, which we summarize below:
>
> * Table 2a: VERA generates significantly more diverse and novel prompts (lower self-BLEU and BLEU-to-template).
> * Table 2b: Baselines perform well only when initialized with strong jailbreak templates; otherwise, their ASR drops sharply. VERA remains stable.
> * Table 2c: Under a fixed time budget, VERA produces far more successful attacks, highlighting its scalability for red-teaming.
>
> **Table 2a: Diversity and Novelty of Attacks (for Fig. 2a)**
> | Metric              | VERA (Ours) | GPTFuzzer | GPTFuzzer * | AutoDAN | AutoDAN * |
> |---------------------|-------------|-----------|-------------|---------|-----------|
> | self_bleu (↓)       | 0.420       | 0.846     | 0.924       | 0.788   | 0.709     |
> | bleu_template (↓)   | 0.003       | 0.450     | 0.337       | 0.649   | 0.241     |
>
> **Table 2b: Fixed Query Budget — 100 Prompts (for Fig. 2b)**
>
> | Metric | VERA (Ours) | GPTFuzzer | GPTFuzzer * | AutoDAN | AutoDAN * |
> |--------|-------------|-----------|-------------|---------|-----------|
> | ASR    | 78.62       | 85.14     | 70.90       | 81.08   | 66.30     |
> | Time   | 697         | 1282      | 1755        | 455     | 1250      |
>
>
>
> **Table 2c: Fixed Time Budget — 1250 Seconds**
>
> | Metric              | VERA (Ours) | GPTFuzzer | GPTFuzzer * | AutoDAN | AutoDAN * |
> |---------------------|-------------|-----------|-------------|---------|-----------|
> | Successful Attacks  | 610         | 83        | 50          | 256     | 66        |
>
>
>
>
> * **(Q3)Figure 2 (b) needs more explanations, especially the vertical lines in the plot, I am not following this part.**
>
> We will improve the visualization and annotations in Figure 2(b) in the revised version to make it more interpretable. Due to the limitations of the rebuttal format, we are including the underlying results as Tables 2a–2c above to clarify what each subfigure demonstrates.
>
> **Interpreting Figure 2(b) and Table 2b – Fixed Query Budget**
>
> * Y-axis: Number of successful attacks.
> * X-axis: Wall-clock time (in seconds)
>
> **Vertical lines**:
> * For VERA: first vertical line = end of the training stage, second vertical line = end of the sampling/attack stage (100 attacks)
> * For baseline methods: single vertical line = point where they reach the query cap (100 attacks)
>
> In Figure 2(b), we compare VERA and GPTFuzzer under a strict budget of 100 attack attempts. GPTFuzzer initially outperforms VERA due to the absence of a training phase and access to strong initial templates. However, when GPTFuzzer is restricted to non-jailbreaking templates (GPTFuzzer*), VERA achieves higher ASR and faster generation, even accounting for training time.
>
> We augment these results with Table 2b, which includes AutoDAN, a strong genetic baseline. Although AutoDAN and GPTFuzzer perform well with strong templates, their advantage vanishes under template constraints, highlighting VERA’s robustness to initialization and the benefits of optimization-based search.
>
> We also evaluate performance under a time budget of 1250 seconds (Table 2c), which better reflects real-world red-teaming. In this setting, VERA’s amortized inference offers a clear advantage: after training, it produces attacks in a single pass, while GPTFuzzer and AutoDAN restart search each time. As a result, VERA discovers significantly more vulnerabilities within the same time frame.
>
> * **(Q4)Please add the simple baselines mentioned earlier.**
>
> We address this in the response to (W1).
>
> * **(Q5)How the baseline results are collected? Is results in table 1 done on the same set of samples?**
>
> Yes, all results in Table 1 are evaluated on the same set of target behaviors, as provided by the HarmBench benchmark and using the same evaluation classifier provided by the HarmBench.
>
> * **(Q6)For well safety-aligned models, any insights on how VERA will perform better?**
>
> Stronger alignment reduces jailbreak success by shifting probability mass away from harmful outputs, but cannot eliminate risk entirely [2]. As long as harmful content exists in training data, some prompts will induce unsafe behavior.
>
> From a distributional standpoint, successful jailbreaks lie within a low-probability subspace of the model’s input–output distribution. Alignment shrinks and shifts this subspace but does not collapse its measure to zero. VERA models a behavior-specific distribution and uses gradient-based variational inference to adaptively explore and recover prompts from this sparse, shifted region.
>
> Our experiments demonstrated that VERA maintains strong performance on safety-aligned models, including LLaMA‑2, R2D2 and LLaMA‑3 (Tables 1 and 3 in the paper). This is because VERA is tailored towards each distribution while maintaining coherent and fluent attacks. While prior black-box baselines are fluent, they rely on the initial templates remaining effective, making them vulnerable to shifts in the alignment distribution. White-box models can adapt to the new distribution, but are often incoherent and thus easy to detect via simple perplexity filters. VERA captures the best of both adaptivity and fluency, making it effective on strongly aligned safety models.
>
> [1]. HarmBench: A Standardized Evaluation Framework for Automated Red Teaming and Robust Refusal. Mazeika, Mantas et al. 2024.
>
> [2]. Mission Impossible: A Statistical Perspective on Jailbreaking LLMs. Su, Jington et al. NeurIPS 2024.
>
> [3]. Ronald J Williams. Simple statistical gradient-following algorithms for connectionist reinforcement learning. 1992.
>
> [4]. REINFORCE v.s Parameterization Trick. Javed, Sayar Ashed. 2018.

---

> > ### Author Response · Authors · 2025-08-06
> >
> > This is a gentle reminder to take a look at our rebuttal. We have addressed your initial concerns in detail, and would appreciate any additional feedback or response before the end of the discussion period.

---

> > > ### Comment · Reviewer_heHX · 2025-08-06
> > > **Thank you for the detailed response**
> > >
> > > Most of my concerns are addressed. Thanks for the detailed responses. I will adjust my score accordingly.

---

### Official Review · Reviewer_XTYn · 2025-07-03

**Clarity:** 3
**Significance:** 3
**Originality:** 3
**Rating:** 5
**Confidence:** 4

**Summary:**

Current jailbreaking methods require white-box access or induce high computational costs due to the searching loops. This paper proposes VERA, an automated black-box attack method that enables efficient and diverse adversarial prompt generation. VERA employs the variational inference framework to learn a language model $q_{\theta}(x)$ generating adversarial prompts that are likely to elicit harmful responses from a target LLM. Extensive empirical results show that VERA achieves competitive performance compared to previous white-box and black-box methods and demonstrates robustness against common defense mechanisms.

**Questions:**

Please see Weaknesses.

**Ethical Concerns:**

["NO or VERY MINOR ethics concerns only"]

**Final Justification:**

Although VERA does not outperform in every case, I believe that this paper offers a new paradigm that can efficiently generate diverse jailbreaking prompts. Therefore, I recommend acceptance.

**Limitations:**

yes

**Quality:**

3

**Strengths And Weaknesses:**

## Strengths
- The idea of using variational inference framework is novel and appealing, different from standard optimization approaches. I really like the way this method formulates and learns the distribution of jailbreak prompts, which inherently offers the diversity of the prompts. I also find the connection to RLHF in A.3 quite interesting.
- The experiments are extensive, covering many LLMs and jailbreaking methods.
- VERA achieves high performance, outperforming black-box attacks and only underperforms white-box attacks on LLama 7/13B.

## Weaknesses
- This approach relies on the judge model, however, its presentation is not clear. If we use the binary classifer, how do we create the dataset to train the judge model? If we prompt another LLM, what is the template?
- The experiments should also explore the performance of VERA with different judge models, e.g., stronger/weaker judge.
- While the attack experiments are quite intensive, the paper only considers two defenses. I'd suggest evaluating on other jailbreaking defenses such as LLamaGuard or SmoothLLM for more comprehensive benchmark.

---

> ### Author Rebuttal · Authors · 2025-07-31
>
> Thank you for the valuable review and for recognizing VERA’s novelty, diversity, and strong experimental coverage. We address your concerns below.
>
> * **(W1)This approach relies on the judge model, however, its presentation is not clear. If we use the binary classifer, how do we create the dataset to train the judge model? If we prompt another LLM, what is the template?**
>
> In our setup, we **do not train a judge model ourselves**. Instead, we use the **pretrained validation classifier** provided by **HarmBench** [1], which outputs a **scalar harmfulness probability** for a given prompt–completion pair. This classifier is trained on a **binary human-annotated dataset** distinguishing harmless vs. harmful completions, and was specifically designed for use inside optimization loops. The validation classifier operates over plain prompt–response pairs and does not require templated inputs. Its architecture and training procedure are described in [1]. While our method is compatible with any scalar-valued judge model, we will explicitly include these details in Section 4.1 in the revised version for clarity.
>
> * **(W2)The experiments should also explore the performance of VERA with different judge models, e.g., stronger/weaker judge.**
>
> We agree with the reviewer that the choice of judge model can influence performance and that evaluating VERA under different judges is important. So, we include an ablation experiment to evaluate VERA using multiple judge models of varying strengths, including the HarmBench validation classifier and another fine-tuned LLM-based classifier [5] and GPT 4o mini judge (with template from [6]). The results are as follows:
>
> **Table 1: Judge Ablation for VERA**
>
> | Judge | GPT-4o mini | SR   | HB   |
> |-------|-------------|------|------|
> | ASR   | 83.0        | 89.0 | 94.0 |
>
> VERA achieves consistent improvement in performance across all judge models, with higher ASR when using a stronger judge (HB). Weaker judges like GPT-4o mini still allow successful optimization but result in lower attack success, due to noisier and less calibrated feedback (higher false positives).
>
> * **(W3)While the attack experiments are quite intensive, the paper only considers two defenses. I'd suggest evaluating on other jailbreaking defenses such as LLamaGuard or SmoothLLM for more comprehensive benchmark.**
>
> Thank you for the question. We include additional results below for LlamaGuard [2], SmoothLLM [3], and Robustly-aligned LLM (RA-LLM) [4]. We compare against GCG and AutoDAN, as these represent strong white-box and black-box baselines. We apply the attackers to the target model, and then apply the defense methods to the generated prompts and measure the by-pass rate.
>
> **Table 2: LLaMA Guard Bypass Rate**
>
> | Method   | Bypass Rate (%) |
> |----------|------------------|
> | **VERA**     | 17.50           |
> | GCG      | 5.06            |
> | AutoDAN  | 6.25            |
>
> On Llama-Guard, we hypothesize that GCG-style suffixes and AutoDAN templates are more easily detected because similar prompts may have been included in its training set, given their widespread use. In contrast, VERA’s behavior-specific attacks are more diverse and less templated, likely falling outside the distribution LLaMA Guard was trained on and enabling higher bypass rates.
>
>
> **Table 3: SmoothLLM (Rejection Phrase Heuristic – Overestimates Bypass)**
>
> | Method         | # Smoothing Copies | Perturbation Type         | Perturbation % | Reported JB % |
> |----------------|---------------------|----------------------------|----------------|----------------|
> | **VERA** | 10                  | RandomSwapPerturbation     | 10             | 58.75          |
> | GCG            | 10                  | RandomSwapPerturbation     | 10             | 41.77          |
> | AutoDAN        | 10                  | RandomSwapPerturbation     | 10             | 95.00          |
>
> On SmoothLLM, character swaps break GCG’s brittle syntax but preserve AutoDAN’s semantic coherence, enabling strong performance. VERA underperforms AutoDAN due to the lack of handcrafted initialization, but outperforms GCG by learning more semantically grounded prompts.
>
>
> **Table 4: RALLM Evaluation (Various Thresholds)**
>
>
> | Threshold Setting              | Method   | Bypass Rate (%) |
> |--------------------------------|----------|------------------|
> | `20,10,0.3,0.2`                | GCG      | 3.79             |
> |                                | AutoDAN  | 0.00             |
> |                                | **VERA**     | 2.50             |
>
> On RA-LLM, token-level removals often break the semantics of AutoDAN’s fixed templates by deleting entire words, while occasionally preserving GCG’s adversarial suffixes, which results in a small number of successful bypasses. Like AutoDAN, VERA is sensitive to such removals, which can break prompt semantics. However, its diverse prompt distribution increases the likelihood that some prompts remain robust, giving it greater resilience than AutoDAN.
>
> Overall, VERA maintains strong and stable performance across all defenses. It outperforms GCG and AutoDAN on LLaMA Guard, remains competitive on SmoothLLM despite lacking handcrafted templates, and shows greater robustness than AutoDAN under RA-LLM due to its diverse prompts.
>
>
> [1]. HarmBench: A Standardized Evaluation Framework for Automated Red Teaming and Robust Refusal. Mazeika, Mantas et al. 2024.
>
> [2]. Llama Guard: LLM-based Input-Output Safeguard for Human-AI Conversations. Inan, Hakan et al. 2023.
>
> [3]. SmoothLLM: Defending Large Language Models Against Jailbreaking Attacks. Robey, Alexander et al. 2024.
>
> [4]. Defending Against Alignment-Breaking Attacks via Robustly Aligned LLM. Cao, Baichuan et al. 2024.
>
> [5]. A strongreject for empty jailbreaks. Souly, Alexandra et al. 2024
>
> [6] AutoDAN-Turbo: A Lifelong Agent for Strategy Self-Exploration to Jailbreak LLMs, Liu, Xiaogeng et al. 2024

---

> > ### Comment · Reviewer_XTYn · 2025-08-06
> >
> > Thank you for your response and discussion on additional results, although VERA does not outperform in every case. Indeed, I do not expect VERA to be the state-of-the-art, which it's not designed to be, but a new paradigm that can efficiently generate diverse jailbreaking prompts. I believe that these results and discussion would strengthen the paper. Therefore, I recommend acceptance.

---

### Official Review · Reviewer_PvsC · 2025-07-03

**Clarity:** 3
**Significance:** 3
**Originality:** 2
**Rating:** 4
**Confidence:** 3

**Summary:**

The paper focuses on automatically generating jailbreak attack prompts by optimizing the attacker model using a variational inference-based framework. It frames jailbreak prompt generation as a posterior inference problem and trains the attacker model with LoRA to learn a distribution over effective jailbreak prompts  targeting harmful behaviors. This approach eliminates the need for genetic algorithms or manually crafted prompts. Experimental results on the HarmBench dataset across different LLMs demonstrate its effectiveness.

**Questions:**

1. Why does VERA achieve better transferability compared to prior optimization-based methods?
2. Table 4 shows performance across different attacker LLMs. Why does Vicuna 7B significantly outperform LLaMA 3 8B? Is there any discussion?

**Ethical Concerns:**

["NO or VERY MINOR ethics concerns only"]

**Final Justification:**

The authors' rebuttal has addressed most of my concerns and I am inclined to keep my original positive rating considering its limitation of using single-sample estimation.

**Limitations:**

Reward shaping or curriculum learning to gradually increase task difficulty or focus training on partial success signals can improve gradient signals\ density.

**Quality:**

3

**Strengths And Weaknesses:**

**Strengths**
1. Effectively generates diverse jailbreak prompts without relying on human annotations, which is applicable to both open-source and closed-source models.
2. Comprehensive experiments and analyses across multiple LLMs.
3. The paper is well-written and presented.

**Weaknesses**
1. The paper lacks justification for why the judge score is a valid approximation for the likelihood of generating harmful responses given specific prompts.
2. No discussion and comparison with RL-based jailbreak attack methods, such as approaches that directly optimize toward successful attacks [1, 2].
3. The BLUE score alone is insufficient for measuring diversity in semantics.
4. The experimental results are limited to a single dataset: HarmBench.
5. Table 2 should include ASR results from baseline attacker models for comparison.

[1] RL-JACK: Reinforcement Learning-powered Black-box Jailbreaking Attack against LLMs
[2] Reinforcement Learning-Driven LLM Agent for Automated Attacks on LLMs

---

> ### Author Rebuttal · Authors · 2025-07-31
>
> Thank you for your helpful feedback. We appreciate your recognition of VERA’s diversity and its comprehensive experiments. We address your concerns below.
>
> * **(W1) The paper lacks justification for why the judge score is a valid approximation for the likelihood of generating harmful responses given specific prompts.**
>
> To clarify, we do not use the judge model to predict the harmfulness of a prompt directly. Instead, we first obtain a response $\hat{y}$ from the target model, and then use a judge to evaluate whether this response is harmful. Thus, the judge's score reflects the harmfulness of a realized response.
> Mathematically, we aim to estimate:
> $$
> P_{LM} (y \in Y_{\text{harm}} | x) = \sum_y P_{LM}(y | x) \mathbf{1}(y \in Y_{\text{harm}}) =  E_{y \sim P_{LM}(\cdot | x)} [ \mathbf{1}(y \in Y_{\text{harm}})]
> $$
> Here, $Y_{\text{harm}}$ represents the set of harmful responses, and $y \in Y_{\text{harm}}$ corresponds to $y^*$ in our submission.  Furthermore, $\mathbf{1}(y \in Y_{\text{harm}})$ represents an indicator function that returns $1$ if $y \in Y_{\text{harm}}$, $0$ otherwise. Directly computing this expectation is intractable. To approximate it, we can use a judge to approximate $J(x, y) \approx \mathbf{1}(y \in Y_{\text{harm}})$. Since multiple samples are expensive, we use a single sample $\hat{y} \sim P_{LM}(\cdot | x)$ and compute:
> $$P_{LM} (y \in Y_{\text{harm}} | x) \approx J(x, \hat{y})$$
>
> This yields a one-sample Monte Carlo estimator. As long as $J(x, y)$ is reasonably accurate in evaluating the harmfulness of responses, it serves as a valid approximation for this quantity. Prior works [1,2,3,4] have also used judge models similarly to guide optimization in black-box settings.
>
>
> * **(W2) No discussion and comparison with RL-based jailbreak attack methods, such as approaches that directly optimize toward successful attacks [1, 2].**
>
> We briefly mention RL Jack [1] in our related works section. To expand further, this method does not train a model to generate jailbreak prompts; it uses RL to decide which genetic transformation to apply to a pre-existing prompt. This optimization operates over discrete mutation operations, similar to the genetic algorithm baselines (e.g., AutoDAN) already included in our comparisons.
>
> Thank you for making us aware of [2]. We highlight some key differences below.
>
> *Reinforcement Learning Targeted Attack (RLTA, Wang et al 2024).*
>
> * Train a single attacker across behaviors
> * Fine-tune the entire attacker LM, resulting in 96 hour overhead
> * Use general-purpose exploits
> * Identify isolated failure modes
>
> *VERA (Ours)*
> * Train an attacker to learn the distribution of behavior-specific adversarial prompts
> * Learn a light-weight LORA Adaptor, allowing for full training in <10 minutes
> * Use diverse, behavior-specific exploits
> * Provide a distributional view of vulnerabilities
>
> Due to the lack of publicly available code or model weights, we are unable to make comparisons against [2].
>
> * **(W3) The BLUE score alone is insufficient for measuring diversity in semantics.**
>
> We agree that BLEU alone does not capture semantic diversity. However, in the context of jailbreak prompts, embedding‑based metrics (e.g., BERTScore) can be misleading because prompts for the same harmful behavior are often semantically similar and share key terms, causing embeddings to overestimate similarity. In contrast, BLEU provides a reliable and interpretable estimate of lexical diversity. A low BLEU score is a necessary condition for diversity, as minimal n‑gram overlap strongly implies non‑duplicate prompts. Prior works on jailbreak prompt generation (e.g., [6]) have similarly adopted BLEU as the diversity metric.
>
> * **(W4) The experimental results are limited to a single dataset: HarmBench.**
> HarmBench is a comprehensive, standardized benchmark explicitly designed for automated LLM red-teaming and jailbreak evaluation. It contains 400 unique harmful behaviors, spanning 7 semantic categories (e.g., cybercrime, harassment, misinformation, chemical/biological weapons, copyright violations) and multiple functional categories (standard, contextual, copyright). This breadth ensures coverage across the most practically relevant malicious-use scenarios. Further, it provides a validation/test split and fine-tuned classifiers, ensuring results are directly comparable across studies. Similar to recent works which only use Harmbench for evaluation [9,10,11].
>
> To broaden evaluation, we also include results on AdvBench from [3]. Following [7] we used 50 most harmful questions due to time limit and keyword based ASR computation as in [7].
> | Method     | Llama2-7b-chat | Vicuna-7b |
> |------------|----------------|-----------|
> | GCG        | 0.10           | 0.72      |
> | AutoDAN    | 0.12           | 0.82      |
> | GPTFuzzer  | 0.12           | 1.00      |
> | PAIR       | 0.08           | 0.64      |
> | VERA       | 0.16           | 0.86      |
>
> These results show that VERA remains competitive and generalizes beyond HarmBench.
>
> * **(W5) Table 2 should include ASR results from baseline attacker models for comparison.**
>
> We would like to clarify what metrics you believe are missing from Table 2. If you are referring to the ASR for the initial model, we omit this information as it is uninformative. We measure transferability by generating prompts for the base target model and filtering based on the success, similar to [4, 5]. These successful attacks are then applied to other models to measure whether successfully jailbreaking the initial model is sufficient to successfully jailbreak a new, unseen model. Thus the ASR on the original model is 100% by default.
>
> * **(Q1) Why does VERA achieve better transferability compared to prior optimization-based methods?**
>
> To clarify, we do not claim that our method produces more transferable attacks when compared to alternative approaches. In Table 2, we only include results for transferability for VERA. We do not compare against other methods as the typical methodology for measuring transferability does not allow for meaningful comparisons between methods [8], as the metric is dependent on how many generated attacks are effective on the original method. Thus an algorithm that produces many successful attacks on the baseline may seem to have worse transferability than an algorithm that produces only a few attacks.
>
> Furthermore, transferability is orthogonal to our primary research focus, as we focus on black-box jailbreaking, there is not necessarily a need for good transferability, as it will always be possible to directly apply VERA to the target model. We include transferability results to highlight that while there is overlap in terms of model vulnerability, this overlap is asymmetrical as attacks against strong models transfer to weaker models but not vice-versa.
>
> * **(Q2)  Table 4 shows performance across different attacker LLMs. Why does Vicuna 7B significantly outperform LLaMA 3 8B? Is there any discussion?**
>
> This is because Vicuna and Llama 3 use different safety alignment techniques, making Vicuna a more effective base for an attack model. We optimize the following objective:
>
> $$E_{q_{\theta}(x)} [ \log P_{LM} (y^* | x) + \log P(x) - \log q_{\theta}(x) ]$$
>
> Here, $\log P_{LM}(y^* | x)$ reflects the jailbreaking effectiveness of the generated prompt $x$, and $\log P(x)$ reflects the probability of the prompt under the original attacker distribution. Both terms are important: the first ensures attack success, and the second ensures fluency of the resulting prompt. The learned parameters must balance both terms: removing the first results in poor ASR, and removing the second makes the attacks easy to defend against using perplexity filters.
>
> However, for attack models that are well aligned with safety standards, the first and second term conflict with each other: the prompts that would have a very high judge score would also be low in probability under the base model due to containing harmful content, and vice-versa. This makes the optimization landscape more difficult to traverse, resulting in lower performance.
>
> In contrast, Vicuna 7B, which is less safety-aligned, provides a more permissive prior: it allows fluent but still harmful prompts to have higher $P(x)$, enabling better optimization of the full objective.
>
> [1]. RL-JACK: Reinforcement Learning-powered Black-box Jailbreaking Attack against LLMs. Chen Xuan et al.
>
> [2]. Reinforcement Learning-Driven LLM Agent for Automated Attacks on LLMs. Wang, Xianweng et al. ACL Proceedings of Workshop in Privacy of Natural Language Processing, 2024.
>
> [3]. Universal and Transferable Adversarial Attacks on Aligned Language Models. Zou, Andy et al. 2023.
>
> [4]. Jailbreaking Black Box Large Language Models in Twenty Queries. Chao, Patrick et al. 2024.
>
> [5]. Tree of Attacks: Jailbreaking Black-Box LLMs Automatically. Mehorotra, Anay et al. 2024.
>
> [6]. COLD-Attack: Jailbreaking LLMs with Stealthiness and Controllability. Guo, Xingang et al. 2024
>
> [7]. When LLM meets DRL: Advancing jailbreaking efficiency via drl-guided search. Chen, Xuan et al. 2024
>
> [8]. AutoDAN-Turbo: A Lifelong Agent for Strategy Self-Exploration to Jailbreak LLMs. Liu, Xiaogeng et al. 2024
>
> [9]. Tail-aware Adversarial Attacks: A Distributional Approach to Efficient LLM Jailbreaking. Beyer, Tim et al. 2025
>
> [10]. LLMStinger: Jailbreaking LLMs using RL fine-tuned LLMs. Jha, Piyush et al. 2024
>
> [11]. On Jailbreaking Quantized Language Models Through Fault Injection Attacks. Zahran, Noureldin et al. 2025

---

> > ### Comment · Reviewer_PvsC · 2025-08-05
> >
> > Thanks for your response which has addressed most of my concerns. However, regarding Weakness 1, while I understand that you’re estimating the harmfulness via a one-sample Monte Carlo approach using the judge model, I’m still unsure whether using this estimator may introduce high variance or bias, especially when the output distribution is diverse and only a single sample is used. It would strengthen the justification if you could provide additional analysis on the variance or robustness of this approximation (e.g., via multiple samples), or clarify under what conditions this one-sample estimator remains reliable.

---

> > > ### Author Response · Authors · 2025-08-06
> > >
> > > Thank you for raising this important point. We agree that a one-sample estimator may, in general, exhibit high variance or bias, particularly when the output distribution of the model is diverse. Below, we clarify both aspects and discuss conditions under which the estimator remains reliable.
> > >
> > > **Bias**
> > >
> > > Since our estimator uses a judge model as a proxy for the indicator $\mathbf{1}(y \in Y_{\text{harm}})$, it could be biased if the judge diverges from the true labeling function. This limitation is unavoidable in black-box settings, as exact likelihoods are inaccessible. The consistent empirical success of VERA across multiple models (Table 1 in the submission) suggests that the judge model is sufficiently unbiased: if the judge systematically misclassified harmful responses, VERA would not reliably discover effective jailbreak prompts. Due to the black-box setting, it is not possible to provide stronger theoretical guarantees about the judge’s accuracy, making empirical performance the most reliable indicator of estimator quality.
> > >
> > >
> > > **Variance**
> > >
> > > The variance of this estimator directly depends on the consistency of the target model’s outputs for a given prompt. If the model produces highly distinct responses for the same prompt, the estimator will exhibit high variance; conversely, if the model consistently generates similar responses, the variance will be low. To analyze the stability of the estimator, we evaluated the standard deviation of judge scores across 10 sampled responses for each of 80 prompts (sampled from Vicuna 7B at temperature 0.7). We report the following:
> > >
> > > | Metric                          | Value  |
> > > |---------------------------------|--------|
> > > | Mean Standard Deviation         | 0.107  |
> > > | Variance of Standard Deviations| 0.012  |
> > >
> > > These values suggest that for a fixed prompt, the variability in the judge’s output across generations is low. This is consistent with the fact that deployed language models are typically configured with low-temperature decoding, which concentrates probability mass on a small subset of outputs, thereby reducing sampling variance.
> > >
> > > In general, the estimator is reliable if the judge accurately approximates the harmfulness indicator (low bias) and the target model produces consistent responses for a given prompt (low variance). Our empirical analysis demonstrates that both conditions hold sufficiently well in practice.

---

> > > > ### Author Response · Authors · 2025-08-07
> > > >
> > > > Thank you again for your thoughtful feedback and engagement during the rebuttal. We wanted to follow up on your most recent comment, where you raised a question regarding the bias and variance of the judge estimator used in VERA. We’ve provided a response addressing this point and wanted to ensure you had a chance to read it before the discussion period ends.

---

> > > > > ### Comment · Reviewer_PvsC · 2025-08-08
> > > > >
> > > > > Thank you for your additional response showing minimal bias and variance of your one-sample estimation. However, I still believe a multiple-sample-based estimator would be more robust given its importance in this work, especially for further applications with large reasoning models, where temperature sampling often performs better. As I initially gave a positive score, I am inclined to keep my rating.

---

> > > > > > ### Author Response · Authors · 2025-08-08
> > > > > >
> > > > > > We appreciate the reviewer’s thoughtful follow-up. We would like to quickly remark that **VERA is not restricted to single-sample estimation** and **is fully compatible with multi-sample estimators**. We chose the single-sample estimation for efficiency and because our empirical results (Table 1 in submission) showed it to be sufficiently stable in practice. Thus, the number of samples is a hyper-parameter can be tuned freely based on scenarios. We thank the reviewer for noting this hyper-parameter and will add a discussion in the appendix.

---

### Official Review · Reviewer_4c2D · 2025-07-03

**Clarity:** 3
**Significance:** 3
**Originality:** 3
**Rating:** 5
**Confidence:** 4

**Summary:**

This paper introduces a new black-box framework to generate jailbreaks based on variational
inference. The framework fine-tunes an attacker model using a LORA adapter optimizing
an objective which optimizes for likelihood of harmful content, fluency of the
jailbreak, and diversity of outputs. The method is tested on HarmBench and evaluated against
a set of baselines showing that it overperforms almost all of them, in almost all scenarios
tested.

**Questions:**

- You claim that the method is "future-proof". In what sense? That it is going to work also
against different alignment techniques? Or that it works also if providers hardcode fixes to
block, e.g., GCG-like suffixes?
- Did you try the attack against more recent models, which seem to be more robust to jailbreaks
than the older models tested in the paper?
- How does the attack compare in terms of efficiency when compared to universal GCG suffixes?
- In my understanding, a LORA adapter correspond to one specific behavior. Do you think it
would be possible to have "unversal" LORA adapters that can work with multiple behaviors?
- How do you make sure the LLM-classifier used in the optimization is well calibrated?

**Ethical Concerns:**

["NO or VERY MINOR ethics concerns only"]

**Final Justification:**

I stand by the strengths I put in my review, and the authors addressed one of the two weaknesses I pointed out, and I believe the second weakness is not enough to not recommend acceptance.

**Limitations:**

Yes.

**Quality:**

3

**Strengths And Weaknesses:**

Strenghts:

- Motivation is good: being able to sample jailbreaks from a distribution
is useful to scale red-teaming up.
- The more mathematical part is very well explained.
- The method is very effective on relatively old models.
- The method is effective against Circuit Breakers, which, to the best of my knowledge,
claim to be unbroken (in token input space).

Weaknesses:

- The models tested are somewhat old. It is unclear if this attack also works against newer
models which might employ more advanced alignment techniques (or reasoning).
- Some core experimental details (e.g., what judge is used) should be in the main part of the
paper.

---

> ### Author Rebuttal · Authors · 2025-07-31
>
> Thank you for your supportive feedback. We appreciate your recognition of our method’s strong motivation, clear mathematical formulation, and effectiveness. We address your comments below.
>
> * **(W1) The models tested are somewhat old. It is unclear if this attack also works against newer models which might employ more advanced alignment techniques (or reasoning).**
>
> We clarify that our experiments already included newer models and stronger alignment techniques. Specifically, R2D2 (2024) [1] is an adversarially trained LLM designed to resist jailbreaks. We focus on the models in Table 2 to enable robust, controlled comparisons via HarmBench, which provides standardized results across multiple attack methods. In Table 3, we provide results for LLaMA 3 8B-Instruct, both with and without the circuit-breaker (2024) [6] defense. This represents a recent, strongly aligned model.
>
> Jailbreaking reasoning models is an exciting future direction but beyond the scope of this work. These models obtain the final answer after a potentially long reasoning chain. Since VERA relies on feedback from the model’s final output, applying it naively would require executing the full chain at each step, making optimization expensive. Recent work often integrates the reasoning process into the attack itself [4, 5]. Extending VERA to reasoning models would require new algorithmic designs that account for the reasoning process, which we are eager to explore in future work.
>
> * **(W2) Some core experimental details (e.g., what judge is used) should be in the main part of the paper.**
>
> We appreciate the reviewer’s comment and will revise the paper to include these core experimental details in the main section. To clarify, we use the **validation classifier** from HarmBench as our judge model. As described in [1], HarmBench provides two classifiers: an **evaluation** classifier and a **validation** classifier, the latter intended for use within optimization loops to produce scalar harmfulness scores while avoiding contamination or overfitting to the test-time evaluation metric. This aligns with our setup, where the judge supplies continuous feedback for updating adversarial prompts. While our method is compatible with any scalar-valued judge model, we will explicitly include these details in Section 4.1 in the revised version for clarity.
>
> * **(Q1) You claim that the method is "future-proof". In what sense? That it is going to work also against different alignment techniques? Or that it works also if providers hardcode fixes to block, e.g., GCG-like suffixes?**
>
> We argue that **VERA is future proof as it is more robust to improvements in alignment and defense techniques** when compared to prior methods. We can regard successful jailbreaks as lying within a low-probability subspace of the model’s input–output distribution. While alignment can shrink and shift this region, **it can never eliminate adversarial attacks completely** —as discussed in [2], adversarial vulnerabilities persist as long as harmful content exists in the training distribution.
>
> Prior methods either **rely on predefined prompt templates** for initialization (e.g., AutoDAN) or contain **easily identifiable artifacts**  (e.g., high entropy suffixes GCG). These are brittle: they can be neutralized by hardcoded filters, memorization, or even simple perplexity-based defenses (see Table 3). They also implicitly assume that the modes of the jailbreak distribution remain near the initial templates—a fragile assumption under distributional shifts caused by stronger alignment. Empirically, as shown in Figures 2(b) and 2(c), these methods **suffer sharp drops in success rate** when template pools are restricted to ineffective prompts.
>
> In contrast, **VERA does not rely on fixed patterns**. It models a distribution over prompts via variational inference and generates new adversarial prompts through sampling. This adaptability enables it to **discover novel modes of the jailbreak distribution** and circumvent defenses aimed at specific patterns. As a result, VERA remains effective even as alignment shifts the landscape of model vulnerabilities.
>
> * **(Q2) Did you try the attack against more recent models, which seem to be more robust to jailbreaks than the older models tested in the paper?**
>
> We address this question under the point labeled (W1).
>
> * **(Q3) How does the attack compare in terms of efficiency when compared to universal GCG suffixes?**
>
> In terms of inference-time cost alone, universal GCG suffixes are cheaper to apply, as they involve only string concatenation and require no per-instance optimization. However, we argue that this efficiency is short-lived.
>
> Effective red-teaming requires comprehensive coverage of behavior-specific vulnerabilities, not just reliance on general-purpose exploits. Universal GCG suffixes are static by design and thus limited in scope—they may exploit broad weaknesses but cannot adapt to alignment-specific defenses or task-dependent failure modes.
>
> Moreover, if we restrict the efficiency comparison to after the GCG optimization step, we implicitly assume that a finite set of fixed suffixes is sufficient to characterize the model’s full vulnerability space. This assumption becomes untenable as models are re-aligned or updated. In practice, once a model changes or new behaviors are targeted, universal GCG suffixes often need to be re-optimized from scratch—a process that is computationally expensive and must be repeated for each new scenario.
>
> In contrast, VERA amortizes its training cost across many downstream uses. Once trained, it can rapidly generate diverse, behavior-specific adversarial prompts without additional per-example optimization. When accounting for the need to adapt over time or scale to new tasks, the **cumulative cost of maintaining GCG attacks can exceed that of VERA**, making VERA more efficient in realistic red-teaming settings.
>
> * **(Q4) In my understanding, a LORA adapter correspond to one specific behavior. Do you think it would be possible to have "unversal" LORA adapters that can work with multiple behaviors?**
>
> Your understanding is correct. A LoRA adapter is trained to model the distribution over prompts that induce a specific behavior. The goal is to allow the adapter to fully capture the vulnerability landscape associated with that behavior, i.e., instead of finding a single exploit, our algorithm discovers diverse behavior-specific exploits.
>
> It is possible to train a universal LoRA. However, encoding multiple behaviors may exceed the capacity of light-weight LORA adapters, thus requiring larger adapters or full-model fine-tuning. For example, [3] fine-tunes the entire attacker model for 96 hours in order to learn a universal attacker. We believe that the focus on behavior-specific vulnerabilities allows for the use of light-weight LORA adapters and quick optimization loops per behavior.
>
> * **(Q5) How do you make sure the LLM-classifier used in the optimization is well calibrated?**
>
> We rely on pretrained LLM-based classifiers as judges. The classifier used during optimization (the HarmBench validation classifier) is still fairly accurate, achieving 88.6% agreement with human annotators. The evaluation classifier used for final scoring achieves 93.2% agreement.
>
> Regarding our algorithm’s sensitivity to calibration: our method uses the raw probability scores output by the judge, so poor calibration could degrade performance. However, we found the existing classifier to be sufficiently reliable for optimization. Finally, we provide an ablation across different judge models, including the HarmBench validation classifier (HB), the StrongReject fine-tuned classifier [7] (SR),  and GPT 4o mini judge (with template from  [8]). The results are as follows:
>
> **Table 1: Judge Ablation for VERA**
>
> | Judge | GPT-4o mini | SR   | HB   |
> |-------|-------------|------|------|
> | ASR   | 83.0        | 89.0 | 94.0 |
>
> VERA achieves consistent improvement in performance across all judge models, with higher ASR when using a stronger judge (HB). Weaker judges like GPT-4o mini still allow successful optimization but result in lower attack success, due to noisier and less calibrated feedback (higher false positives).
>
> [1]. HarmBench: A Standardized Evaluation Framework for Automated Red Teaming and Robust Refusal. Mazeika, Mantas et al. 2024.
>
> [2]. Mission Impossible: A Statistical Perspective on Jailbreaking LLMs. Su, Jington et al. NeurIPS 2024.
>
> [3]. Reinforcement Learning-Driven LLM Agent for Automated Attacks on LLMs. Wang, Xianweng et al. ACL Proceedings of Workshop in Privacy of Natural Language Processing, 2024.
>
> [4]. A Mousetrap: Fooling Large Reasoning Models for Jailbreak with Chain of Iterative Chaos. Yao et al. 2025.
>
> [5]. AutoRAN: Weak-to-Strong Jailbreaking of Large Reasoning Models. Liang et al. 2025.
>
> [6]. Improving alignment and robustness with circuit breakers. Zou, Andy et al. 2024
>
> [7]. A strongreject for empty jailbreaks. Souly, Alexandra et al. 2024
>
> [8]. AutoDAN-Turbo: A Lifelong Agent for Strategy Self-Exploration to Jailbreak LLMs, Liu, Xiaogeng et al. 2024

---

> > ### Comment · Reviewer_4c2D · 2025-08-02
> >
> > Thank you for your rebuttal and for addressing W2 and the questions I had.
> >
> > However, I believe there is a bit of a contradiction: you claim that the attack is future proof, yet you also mention that in order to work against reasoning models VERA would need to be adapted. Let me be clear, I believe there is still value in an attack which does not work against the most recent, reasoning models, especially because it works also against strong defenses such as circuit breakers. However, the "future-proof" claim seems a but too strong to me.

---

> > > ### Author Response · Authors · 2025-08-03
> > >
> > > Thank you for the follow-up and for pointing this out. We agree that the term "future-proof" may be too strongly phrased. Our intent was to convey that VERA is relatively robust to evolving alignment and defense strategies compared to existing approaches, and that its methodology can be extended to future models, including reasoning models, though some adaptations may be necessary. In the revision, we will remove the term "future-proof" and adjust the phrasing accordingly. We appreciate your thoughtful feedback.

---

> > > > ### Comment · Reviewer_4c2D · 2025-08-04
> > > >
> > > > Thank you for your reply and for the proposed change.

---

### Decision · Program_Chairs · 2025-09-17

**Decision:**

Accept (poster)

**Comment:**

This paper introduces VERA, a variational inference framework for generating jailbreak prompts in the black-box setting. Rather than relying on genetic algorithms or handcrafted templates, VERA trains a lightweight attacker model with LoRA adapters to approximate a distribution over adversarial prompts. Once trained, the attacker can sample fluent, diverse jailbreaks without repeated optimization. The paper presents extensive experiments on HarmBench, along with comparisons against both white-box and black-box baselines, and shows robustness against defenses such as Circuit Breakers, LLaMA Guard, and SmoothLLM.

Strengths:

The core idea of casting black-box jailbreaking as a variational inference problem is original and clearly explained, representing a departure from search-based approaches.

The method is efficient in practice, as the training amortizes across many prompts, and enables diverse sampling.

Experimental coverage is broad, including ablations, multiple attacker backbones, transferability studies, and robustness to defenses.

The rebuttal addressed key reviewer concerns with additional baselines (system-prompt only), judge model ablations, variance analysis of the one-sample estimator, and evaluation against additional defenses.

Weaknesses:

While the evaluation is extensive, most experiments remain centered on HarmBench, limiting the scope of generalization.

The reliance on a judge model introduces possible bias and variance, though the authors provided analysis showing stability in practice.

The performance on strongly safety-aligned reasoning models is less clear; adapting VERA to such settings will require further development.

Some claims (e.g., “future-proof”) were overstated, but the authors have revised their phrasing.


There is strong consensus among reviewers that this is a technically solid and well-presented paper, introducing a new paradigm for black-box jailbreak generation. While not flawless in evaluation breadth, the contributions are clear, timely, and impactful for the area of LLM safety and adversarial robustness. The rebuttal satisfactorily addressed most substantive concerns.